# Antenna Arrangement in UWB Helmet Brain Applicators for Deep Microwave Hyperthermia

**DOI:** 10.3390/cancers15051447

**Published:** 2023-02-24

**Authors:** Massimiliano Zanoli, Erika Ek, Hana Dobšíček Trefná

**Affiliations:** Department of Electrical Engineering, Chalmers University of Technology, 412 96 Gothenburg, Sweden

**Keywords:** deep hyperthermia, cancer treatment, brain tumor, ultra-wide-band, microwave applicator, medulloblastoma, thermal therapy

## Abstract

**Simple Summary:**

Despite the promising results of earlier studies on glioblastomas, hyperthermia is currently not applied in the treatment of brain cancer. Focused intracranial heating is a challenging task due to the presence of critical organs and their extra sensitivity to elevated temperatures. In this contribution, we introduce a new concept to design UWB applicators to achieve adequate temperatures in large brain tumors while protecting the healthy tissues from overheating. We introduce a fast E-field approximation scheme to quickly explore a large number of array configurations to determine the most optimal antenna arrangement around the head with respect to the multiple objectives and requirements of clinical hyperthermia. The proposed solution manages to achieve the level of tumor coverage and hot-spot suppression that is necessary for a successful treatment. The results show that the method is accurate enough to provide qualitative indications about the most suitable antenna arrangement for a given tumor shape and location, while yielding higher target temperatures than annular antenna arrays.

**Abstract:**

Deep microwave hyperthermia applicators are typically designed as narrow-band conformal antenna arrays with equally spaced elements, arranged in one or more rings. This solution, while adequate for most body regions, might be sub-optimal for brain treatments. The introduction of ultra-wide-band semi-spherical applicators, with elements arranged around the head and not necessarily aligned, has the potential to enhance the selective thermal dose delivery in this challenging anatomical region. However, the additional degrees of freedom in this design make the problem non-trivial. We address this by treating the antenna arrangement as a global SAR-based optimization process aiming at maximizing target coverage and hot-spot suppression in a given patient. To enable the quick evaluation of a certain arrangement, we propose a novel E-field interpolation technique which calculates the field generated by an antenna at any location around the scalp from a limited number of initial simulations. We evaluate the approximation error against full array simulations. We demonstrate the design technique in the optimization of a helmet applicator for the treatment of a medulloblastoma in a paediatric patient. The optimized applicator achieves 0.3 ${}^{{}^{\circ}}$C higher T90 than a conventional ring applicator with the same number of elements.

## 1. Introduction

Local hyperthermia for cancer treatment consists of the selective increase in the tumor temperature to ≈40–44 ${}^{{}^{\circ}}$C for about an hour [1]. In combination with radio- or chemo-therapy, this modality has been shown to enhance the therapeutic outcome for several tumor types in clinical trials [2,3,4]. Conformal phased arrays are used in microwave (MW) hyperthermia (HT) to non-invasively deliver the prescribed thermal dose to deep-seated tumors [5]. In this process, it is of paramount importance to subject the target volume to a high and uniform temperature increase while keeping the surrounding healthy tissues within physiologically tolerated temperatures [6].

External MW-HT has been successfully applied to targets in the pelvis and the head and neck with remarkable results. To date, however, no clinical applications in the treatment of brain tumors have been reported, despite early encouraging results obtained with interstitial techniques [7]. The implementation of MW-HT for the treatment of solid brain tumors could be particularly beneficial in paediatric patients, where the incidence of such malignancies is the highest [8]. Current treatment modalities based on chemoradiotherapy are known to cause long-term disorders in survivors [9]. There is, thus, a strong motivation for the development of brain applicators and the introduction of hyperthermia as a means of lowering the ionizing dose while maintaining the same clinical output.

Local heating of tissues in the head is a challenging task due to the presence of critical organs and their extra sensitivity to hyperthermic temperatures [10]. Ideally, the therapeutic range of 40–44 ${}^{{}^{\circ}}$C should be reached everywhere in the tumor, while healthy tissues should not exceed $42{\;}^{{}^{\circ}}$C. Particular care should be devoted to avoiding MW radiation in the eyes [11]. Unfortunately, radio-frequency (RF) waves in the MW range are known to be easily absorbed by biological tissues [12], resulting in poor penetration depth. This is especially true for the cerebrospinal fluid (CSF), due to its high conductivity at these frequencies [13]. The enclosure of the skull (cortical bone) adds to the complication as its dielectric contrast causes irregular wave scattering and multiple reflections. For these reasons, additional efforts must be spent in ensuring that the applicator can reliably target the tumor while minimizing losses in healthy tissue. The latter may result in the formation of hot-spots, which are known to be the limiting factor for the maximum achieved tumor temperature during a treatment session [14].

In a typical MW-HT applicator design, the array is a conformal ring of equally spaced antennas immersed in a water bolus, which fills the gap between the antennas and the patient’s skin. The bolus realizes a dielectric match for an increased power transfer to the body and simultaneously cools off the first layer of tissue where the electromagnetic losses are the strongest [15]. Several groups in the past decades have investigated the relationship between the array design parameters and the resulting ability of the applicator to selectively heat tumors in the pelvis and the neck. These include: operating frequency, array topology (usually ring), distance between antennas, and number of antennas and their distance from the body [16,17,18,19,20]. For brain tumors, external MW-HT has not yet been clinically tested, and the few available non-invasive heating solutions rely on magnetic nano-particles or focused ultra-sound [21]. More recently, however, researchers have begun investigating the feasibility of MW-HT in this anatomical region [22,23,24]. Preliminary results suggest that high-quality heating can be better achieved when the array configuration is customized to the specific tumor location, shape and size [25]. By means of radio-biological modeling, the addition of hyperthermia to the treatment of medulloblastoma has been shown to yield a considerable theoretical boost in the biologically equivalent dose (BED) when combined with stereotactic radiosurgery [26].

In this work, we attempt to go beyond the classical single-frequency ring array configuration and exploit the spherical morphology of the head to develop an ultra-wide-band helmet applicator (250–500 MHz). In doing so, we relax the constraints of fixed distance between the antennas and their mutual alignment. We treat the antenna arrangement around the surface of the scalp as a global optimization problem where each element’s location is left as a degree of freedom. At each iteration of the optimization algorithm, we determine the E-field due to each antenna in the array as the interpolation of a grid of simulated individual antennas at fixed locations. As cost-function for the assessment of a particular array configuration, we utilize a novel metric, the hot-to-cold spot quotient (HCQ), which is based on the specific absorption rate (SAR) distribution and has been shown to correlate well with the resulting temperature increase in deep-seated targets [27,28]. We demonstrate the procedure in the design of several helmet applicators of increasing numbers of elements for the treatment of a paediatric patient with medulloblastoma. We assess the quality of the interpolated field by analyzing the approximation error when compared to an actual simulation. Finally, we quantitatively compare the optimized, semi-spherical arrays to classical elliptical designs of the same number of elements, by developing full thermal treatment plans for each solution.

## 2. Method

### 2.1. Patient Model

We consider a 13-years old male patient with a 126 mL medulloblastoma in the dorsal area of the brain, shown in Figure 1. The tumor is relatively large and extends from the medulla to the skull. The challenge in this patient is due to the hyperthermia target volume (HTV) presenting both peripheral and deep regions, with the distance from the skin surface ranging from ≈1 cm to almost ≈9 cm. The model was obtained via MRI scans with 1mm resolution. The raw data was manually segmented by a trained oncologist into 10 distinct tissues: skin, muscle, bone (cortical), pharynx, cerebrospinal fluid, brain (gray matter), brain (white matter), eye (vitreous humor), cartilage, and tumor. The caudal part of the model, below the brain stem, is filled with muscle to emulate the presence of the rest of the body and allow the simulated wave to propagate with the expected negligible reflection, while reducing the segmentation complexity.

### 2.2. Antenna and Bolus Design

The array elements utilized in our applicator design are self-grounded bow-tie (SGBT) antennas [29], Figure 2. The geometrical parameters of the antennas are optimized to obtain a stable impedance, radiation pattern, and return loss above 10dB across the whole 250–500 MHz band when positioned at a distance of ≈5 cm from the head (measured at the antenna ground plate). This distance is chosen as a compromise between reducing the sensitivity of the antenna response to variations in the patient anatomy (lower at longer distances), and decreasing the losses in the water bolus (lower at shorter distances). The water bolus shape is, thus, obtained by fitting an ellipsoid over a cloud of points randomly located around the scalp and offset by ≈5 cm, as shown in Figure 3. The resulting ellipsoid has a different radius in each direction: 12.5cm along the *x* axis, 14.2cm along *y*, and 14.4cm along *z*. The ellipsoid is trimmed just above the nostrils to provide an opening for breathing, with the cutting plane perpendicular to the *z* axis and being located 7.7cm caudal to the ellipsoid center. Each antenna was placed with its background plate lying as far as possible from the patient while preventing the metal from protruding out of the water.

### 2.3. Numerical Simulations

Electromagnetic (EM) simulations were performed in COMSOL Multiphysics^®^ 5.6 [30]. To reduce the computational burden for the simulation of the interpolation grid (Section 2.5), the patient model was down-sampled to 4mm using a winner-takes-all strategy [31]. This corresponds to approximately λ/18 in the tissue with highest permittivity (CSF), where λ is the wavelength at the highest considered frequency (500MHz). A regular hexahedral mesh was assembled in the patient respecting this step, while the water bolus and the surrounding air background were discretized with a tetrahedral mesh whose resolution varies from λ/30 at the antenna feed and metal corners to λ/5 in the bulk. A convergence test based on the single antenna response (S11) in water was performed to confirm the adequacy of the mesh. The antennas were modelled as sheets of perfect electric conductor (PEC) and excited via a TEM port whose real characteristic impedance was set to the value that minimizes the individual antenna reflection across the bandwidth (26Ω for the SGBT model used in this study). Absorbing conditions (perfectly matched layer, PML) were defined at the domain boundaries. Dispersive healthy tissue properties were retrieved from the IT’IS database [13]. Dispersive tumor properties were obtained as an average of all malignant-tissue properties reported in [32], as recommended by [33].

Thermal (TH) simulations were also performed in COMSOL. The steady-state temperature distribution was determined for each final applicator design. The patient model was added and meshed in the same fashion as for the EM simulation. We added heat-flux boundary conditions to model the convective extraction of heat at the interface between patient and air or water. The chosen convection coefficient between skin and air was 8 W/m${}^{2}$/K [34], while the coefficient between skin and water was 100 W/m${}^{2}$/K [15]. The air temperature was set to $20{\;}^{{}^{\circ}}$C. Due to the proximity of the tumor to the surface, the water bolus directly affects the temperatures in the target volume [15]. Therefore, the water temperature was set to a higher $30{\;}^{{}^{\circ}}$C. Thermal properties were once more obtained from the IT’IS database for each healthy tissue, while the following properties were used for the tumor [33]: density ρ = 1090 kg/m${}^{3}$, specific heat capacity $c_{p}$ = 3421 J/kg/K, and thermal conductivity κ = 0.49 W/m/K.

In the TH simulation, the EM losses were added as a distributed heat source term in the bio-heat equation [35]. This term was obtained from the array’s E-field distribution at each frequency ($E_{f}$) as shaped by the treatment-planning optimization stage (Section 2.4) and obtained by a full array simulation (no interpolation involved):(1)$$\mathrm{PLD}=\kappa \sum_{f}\frac{1}{2}\sigma_{f}{\left|E_{f}\right|}^{2}$$
where PLD stands for power loss density (W/m${}^{3})$ and κ is a scaling factor. Note that, unlike the SAR distribution in Section 2.4, the PLD distribution is not smoothed out nor masked. The value of κ was determined by a local gradient descent optimization whose goal is to obtain a maximum temperature in the healthy tissue equal to $42{\;}^{{}^{\circ}}$C, to respect the toxicity limits in the central nervous system [10]. The resulting temperature distribution in the target volume was assessed by means of the $T_{50}$ and $T_{90}$ indexes [36], i.e., the minimum temperature achieved within the highest 50% and 90% of the temperature distribution in the target, respectively.

### 2.4. Treatment Planning

For each applicator configuration, either during the optimization stage or for final validation, a full multi-frequency SAR-based treatment plan optimization was carried out. The plans were prepared considering the [250,375,500]MHz set of frequencies for simultaneous operation. The optimization variables were the phase and amplitude of each array channel and for each frequency, for a total of $2\cdot n_{f}\cdot n_{c}$ degrees of freedom, where $n_{f}$ is the number of frequencies and $n_{c}$ is the number of channels (antennas). The cost function and goal to be minimized is the hot-to-cold spot quotient (HCQ), defined as follows [27,28]:(2)$${\mathrm{HCQ}}_{p}=\frac{{\bar{\mathrm{SAR}}}_{Rq}}{{\bar{\mathrm{SAR}}}_{Tp}}.$$
where ${\bar{\mathrm{SAR}}}_{Tp}$ is the average SAR in the lowest *p* percentile of target (tumor) tissue, while ${\bar{\mathrm{SAR}}}_{Rq}$ is the average SAR in the highest *q* percentile of remaining (healthy) tissue. The relationship between percentiles is fixed: (3)$$q=p\frac{┌T┘}{┌R┘}$$
where ^┌^⋄_┘_ denotes the volume of the argument. A target percentile *p* of 50% was selected to promote coverage even in the deepest parts of the tumor and increase the resulting temperature indexes. For the present patient model, the corresponding percentile of remaining healthy tissue becomes q=2.8%.

The procedure was implemented in MATLAB^®^ R2021a [37] using our previously devised scheme for the fast minimization of HCQ in multi-frequency problems [38], which is based on an iterative form of time reversal. When a full array simulation is performed on COMSOL, the E-field distributions due to each antenna are directly exported from the software and re-sampled to a uniform matrix with 4mm spatial resolution and single precision. During the array optimization, on the other hand, the individual E-fields were determined by linear interpolation, as described in Section 2.5. The SAR distribution, in W/kg, upon which Equation (Equation 2) has to be evaluated was determined by superposition of the frequency contributions:(4)$$\mathrm{SAR}=\sum_{f}\frac{1}{2}\frac{\sigma_{f}}{\rho }{\left|\sum_{c}\chi_{f,c}E_{f,c}\right|}^{2}$$
where σ is the local material conductivity and ρ its density, while $\chi_{f,c}$ and $E_{f,c}$ are the complex steering parameter and E-field distribution of channel *c* at frequency *f*, respectively.

The SAR was further processed to increase its correlation with temperature. First, the distribution was smoothed out by a 5 g mass averaging scheme within the patient, where surface voxels were treated by expanding the convolution kernel until the mass of tissue within reached 5 g [39]. Secondly, the voxels belonging to the first 20mm of healthy tissue at the surface in contact with the water bolus were completely excluded from the patient mask for the evaluation of the cost-function. This step was included to model the cooling effect of the water bolus in SAR, as the EM losses are effectively counteracted by the convective heat extraction [15]. Additionally, the exclusion of such a thick layer of patient surface was motivated by the knowledge that the most prominent hot-spot was expected to arise in the deep-seated pocket of CSF caudal to the target volume [26], while the peripheral *strati* of CSF were kept within safe temperatures by the joint action of the water bolus and the naturally high perfusion rate of gray matter [13]. Altogether, these measures ensured a high degree of correlation between the SAR and the resulting temperature distribution. All parallel SAR calculations were performed in single precision on a GPU (nVidia^®^ RTX^™^ A6000).

### 2.5. Field Interpolation

To determine the E-field distribution due to a single antenna at any location across the surface of the helmet, we introduced a linear interpolation scheme which relies on a limited number of pre-simulated locations around the head (grid). The procedure consisted of several steps and made use of a local 2D spherical coordinate system (θ,ϕ) mapping the surface of the water bolus, illustrated in Figure 4.

#### 2.5.1. Interpolation Grid

Given the fitted bolus ellipsoid obtained in Section 2.2, a number of points $n_{p}$ were randomly placed around its available surface. The superficial coordinates of each point (θ and ϕ in terms of a local spherical coordinate system aligned with the ellipsoid) were then fed to a local least-squares minimization algorithm (lsqnonlin) which aims at minimizing the sum of the inverse of the squared distances between each pair of points (emulating the repulsion of charged particles of the same sign). Constraints to this optimization stage were the bolus limits, i.e., $\theta_{\mathrm{MAX}}$ in case of a truncated ellipsoidal helmet. The procedure was repeated for increasing $n_{p}$ until the maximum distance between any pair of nearby points fell below a certain target sampling distance. In the patient model at hand, we prepared a grid of $n_{p}=221$ points resulting in a maximum distance of 2.9cm, which is slightly below a half of the minimum wavelength in water (6.8cm @ 500MHz) to provide adequate sampling resolution. The full grid is shown in Figure 5.

For each grid point, a local antenna coordinate system was generated. The origin O=(x,y,z) was initially placed at the surface point corresponding to the spherical coordinates (θ, ϕ) of this grid point. Indicating with *U* the antenna’s orientation (polarization axis), with *W* its main directivity axis (pointing direction), and with *V* a third axis which completes a right-handed (U,V,W) triple, the local coordinate system was obtained by making *W* inwards perpendicular to the ellipsoid’s surface at the point location and finding *U* as the vector tangent to the bolus surface and lying on the ZW plane, where *Z* is the patient’s cranial–caudal axis. Finally, the origin *O* was translated towards the positive *W* direction by the distance necessary to prevent the antenna’s back plate from projecting out of the water ellipsoid. In COMSOL, $n_{p}\cdot n_{f}$ full EM simulations were performed, each with a single antenna model rigidly transformed to match the coordinate system previously prepared. The E-field distributions relative to the individual frequencies were then exported to MATLAB and uniformly re-sampled.

#### 2.5.2. Linear Interpolation

Once the grid distributions were available, the E-field due to a single antenna *a* at arbitrary coordinates $\left(\theta_{a},\phi_{a}\right)\equiv \left(x_{a},y_{a},z_{a}\right)=O_{a}$ could be obtained using a linear interpolation of the distributions relative to the 3 closest grid points $O_{1},O_{2},O_{3}$ (triangular patch), as illustrated in Figure 6:1.A local coordinate system ${\left(U,V,W\right)}_{a}$ was built for the antenna, in a similar way to for the grid points in Section 2.5.1.2.The complex vector E-field distribution $E_{1}$ of the first grid point at frequency *f* was divided everywhere by the local impedance $\eta_{f}$ of the material, yielding a surrogate $H_{1}$ of the H-field of an antenna at that location. This important step was included to render the field distribution less dependent on the patient’s anatomy, thanks to the biological tissues being predominantly non-magnetic.3.This complex vector H-field distribution was transformed to ${\hat{H}}_{1}$ according to a translation $T^{\prime }$, a rotation R, and a second translation $T^{\prime \prime }$, such that:
(5)$$\begin{array}{cc} T^{\prime }\left[O_{1}\right] & =(0,0,0)\\ R[{\left(U,V,W\right)}_{1}] & ={\left(U,V,W\right)}_{a}\\ T^{\prime \prime }\left[\left(0,0,0\right)\right] & =O_{a} \end{array}$$4.The transformed H-field distribution ${\hat{H}}_{1}$ was multiplied by the material impedance $\eta_{f}$ to restore the transformed E-field intensity ${\hat{E}}_{1}$.5.Steps 2 to 4 were repeated for each of the 3 closest grid points.6.The E-field distribution relative to the individual antenna was obtained as a weighed average of the transformed distributions. The weights $\omega_{1},\omega_{2},\omega_{3}$ were determined as the ratio between the area of the subtended triangle to the area of the interpolation patch:
(6)$$\begin{array}{cc} E_{a} & =\omega_{1}{\hat{E}}_{1}+\omega_{2}{\hat{E}}_{2}+\omega_{3}{\hat{E}}_{3}\\ \omega_{1} & =└(O_{a},O_{2},O_{3})┐\;/\;└(O_{1},O_{2},O_{3})┐\\ \omega_{2} & =└(O_{1},O_{a},O_{3})┐\;/\;└(O_{1},O_{2},O_{3})┐\\ \omega_{3} & =└(O_{1},O_{2},O_{a})┐\;/\;└(O_{1},O_{2},O_{3})┐ \end{array}$$
where _└_⋄^┐^ denotes the area of the argument.

#### 2.5.3. Coupling Modeling

The above procedure provides a rough approximation of the E-field of a single antenna in a particular position across the water bolus surface. In any array configuration with two or more antennas, however, coupling phenomena affect the E-field distribution of the single element. We tackled this by utilizing the very individual fields of each antenna to model the coupling distortion of each array element.

To this end, we prepared a separate simulation where a spherical brain phantom is enclosed in a spherical water bolus of the same thickness of our applicator design (≈5 cm), Figure 7a. The phantom includes the same tissues found in the upper hemisphere of the head: brain, cerebrospinal fluid, cortical bone, skin. These were modelled as concentric shells whose thickness was determined by averaging a number of radial samples taken from the patient model, Figure 1. The result was 6.3mm for the skin, 6.8mm for the bone, and 10.7mm for the cerebrospinal fluid. The outer radius of the phantom was 96.9mm, determined in a similar way (average head radius). The inner core is filled with brain material.

Using this model, we determined the coupling factor between two antennas located anywhere inside the bolus. We added a fixed active antenna (A) and generated a number of random locations for a passive antenna (P), including random rotations of its polarization axis, Figure 7b. For each arrangement of this pair, we simulated the individual E-field distributions $E_{A}$ and $E_{P}$ generated when the other antenna is absent, and we extracted the value of $E_{A}$ at the phase center of the passive antenna, $E_{A}\left(O_{P}^{\prime }\right)$. For our SGBT antennas, the phase center $O^{\prime }=O+W\cdot 1.4\;$cm lies in between the flaps, at the end of the feed line, Figure 7c. We projected this value onto the polarization axis of the passive antenna, $U_{P}$, to obtain the complex scalar:(7)$$e_{AP}=\langle U_{P},E_{A}\left(O_{P}^{\prime }\right)\rangle$$
where 〈⋄,⋄〉 denotes the scalar product. Subsequently, we simulated the E-field distribution $E_{A+P}$ due to the active antenna *A* when the passive antenna *P* is present. The passive antenna was terminated by an absorbing TEM port of the same (real) impedance used in excitation mode. Since both antennas are perfect conductors, the overall field is an infinite sum of reflections between the active and passive elements:(8)$$E_{A+P}=E_{A}+k_{AP}\cdot \left(E_{P}+k_{PA}\cdot \left(E_{A}+\ldots \right)\right)$$
where $k_{AP}=k_{PA}$ is the coupling factor between *A* and *P*. Due to losses in the domain, wave propagation and antenna misalignment, the coefficients were expected to be very small. Therefore, one can approximate the overall field as the sum of the impinging field and the first reflection only:(9)$$E_{A+P}\approx E_{A}+k_{AP}\cdot E_{P}$$

From this relationship, the coupling factor $k_{AP}$ can be determined as the spatial average of the ratio between the remainder $E_{A+P}-E_{A}$ and the coupled field $E_{P}$. A more robust fit, however, can be obtained by means of decorrelation:(10)$$k_{AP}\approx \frac{\int_{M}\langle {\bar{E}}_{P},\left(E_{A+P}-E_{A}\right)\rangle \;dM}{\int_{M}\langle {\bar{E}}_{P},E_{P}\rangle \;dM}$$
where M is the domain of the model, i.e., the bolus sphere including the phantom, and $\bar{E}$ denotes the complex conjugate of *E*. Once $e_{AP}$ and $k_{AP}$ have been determined for different arrangements of *A* and *P*, one can study the correlation between the two. For the present study, we generated 30 random pairs and obtained the complex scatter plots shown in Figure 8. The plots confirm the straightforward linear relationship between $e_{AP}$ and $k_{AP}$. A complex coefficient *c* can be fitted on this set of points such that:(11)$$k_{AP}=c\cdot e_{AP}$$
for any arbitrary arrangement of *A* and *P* along the boundary of the water bolus.

This important result enables the calculation of the overall field of an antenna in any array configuration given the individual fields *E* of the single antennas approximated in Section 2.5. For an array of *n* elements, the approximated true field distributions $\hat{E}$ relative to each antenna can be found as:(12)$$\left[\begin{array}{c} {\hat{E}}_{1}\\ {\hat{E}}_{2}\\ \vdots \\ {\hat{E}}_{n} \end{array}\right]={\left[\begin{array}{cccc} 1 & ce_{12} & \cdots & ce_{1n}\\ ce_{21} & 1 & \cdots & ce_{2n}\\ \vdots & \vdots & \ddots & \vdots \\ ce_{n1} & ce_{n2} & \cdots & 1 \end{array}\right]}^{\left(K-1\right)}\cdot \left[\begin{array}{c} E_{1}\\ E_{2}\\ \vdots \\ E_{n} \end{array}\right]$$
where *K* is the total number of wave propagations that should be accounted for (first excitation followed by K−1 reflections). As shown later in Section 3, a sufficient number of propagations is K=3, and throughout the rest of the article we present results obtained utilizing this value.

### 2.6. Approximation Analysis

We quantitatively assessed the approximation error of a single antenna field by comparing the interpolated distribution to an equivalent full simulation in COMSOL. The comparison was carried out for a series of 5 locations within the largest interpolation patch and of increasing distance from a simulated grid point, as shown in Figure 9. We assessed four different aspects of the average relative error between the simulated (SIM) and interpolated (INT) complex vector E-fields: the distribution (DIS), the amplitude (ABS), the phase (ANG), and the direction (DIR). These were calculated as: (13)$$\begin{array}{cc} \epsilon_{\mathrm{DIS}} & =\int_{M}\frac{|E^{\mathrm{SIM}}-E^{\mathrm{INT}}|}{|E^{\mathrm{SIM}}|}dM\;/\;┌M┘\\ \epsilon_{\mathrm{ABS}} & =\int_{M}\frac{\left|\;\right|\langle U,E^{\mathrm{SIM}}\rangle \left|-\right|\langle U,E^{\mathrm{INT}}\rangle \left|\;\right|}{\left|\langle U,E^{\mathrm{SIM}}\rangle \right|}dM\;/\;┌M┘\\ \epsilon_{\mathrm{ANG}} & =\int_{M}\frac{\left|\mathrm{wrap}(∠\langle U,E^{\mathrm{SIM}}\rangle -∠\langle U,E^{\mathrm{INT}}\rangle )\right|}{\pi }dM\;/\;┌M┘\\ \epsilon_{\mathrm{DIR}} & =\int_{M}\frac{\mathrm{acos}(\langle |E^{\mathrm{SIM}}\left|,\right|E^{\mathrm{INT}}|\rangle /\left|\right|E^{\mathrm{SIM}}\left|\right|/\left|\right|E^{\mathrm{INT}}\left||\right)}{\pi /2}dM\;/\;┌M┘ \end{array}$$
where M denotes the patient model volume excluding the first 20mm of tissue in contact with the water bolus, and *U* is the (unitary) polarization vector of the antenna. The SIM and INT distributions were preliminarily mass-averaged according to the scheme described in Section 2.4. The values were evaluated for each individual frequency in the operating set.

### 2.7. Array Optimization

The optimization task must determine the location of each antenna in an array of a given size (number of elements or channels, $n_{c}$). The solver must also make sure that the solution represents a physically feasible arrangement. In particular, the antennas must be placed within the bolus boundaries and they must not overlap with each other. The first requirement is met by providing lower and upper boundaries to the $\theta_{a}$ and $\phi_{a}$ coordinates of each antenna. In the present case, $\phi_{a}$ was unbounded since the ellipsoid covers a full $360^{{}^{\circ}}$ on the XY plane. The second requirement can be implemented as a set of non-linear constraints. If *r* is the radius of the smallest circle enclosing the antenna on its local UV plane, then the following has to be true for any pair (i,j) of antennas:(14)$$\begin{array}{cc} \left|L\right|-l_{i}-l_{j} & >0\\ L & =O_{i}-O_{j}\\ l_{i} & =\sqrt{{\left(r\frac{\langle U_{i},L\rangle }{\left|L\right|}\right)}^{2}+{\left(r\frac{\langle V_{i},L\rangle }{\left|L\right|}\right)}^{2}}\\ l_{j} & =\sqrt{{\left(r\frac{\langle U_{j},L\rangle }{\left|L\right|}\right)}^{2}+{\left(r\frac{\langle V_{j},L\rangle }{\left|L\right|}\right)}^{2}} \end{array}$$
where *L* is the vector from antenna *j* to antenna *i*. This constraint is sufficient as long as the curvature of the bolus ellipsoid is large compared to the size of the antenna along its *W* axis. Further constraints relevant for the HT treatment are the locations of the eyes. The optimizer should not place any antenna in front of these organs as they can be easily damaged by MW radiation. We determined the center *O* and radius *r* of each eye in the model as projected on the water bolus surface, and appended these terms to the set of constraints that was assembled according to Equation (Equation 14).

If the pair $\left(\theta_{a},\phi_{a}\right)$ describes one antenna *a*, then the design procedure must solve a minimization problem consisting of $2\cdot n_{c}$ degrees of freedom. These degrees of freedom, however, are not truly independent from each other. For instance, in the case of 3 antennas, the solution vector:(15)$$\left[\left(\theta_{1},\phi_{1}\right)\left(\theta_{2},\phi_{2}\right)\left(\theta_{3},\phi_{3}\right)\right]$$
represents an array arrangement that is identical to:(16)$$\left[\left(\theta_{2},\phi_{2}\right)\left(\theta_{1},\phi_{1}\right)\left(\theta_{3},\phi_{3}\right)\right]$$
and similar permutations. In other words, there exists a semantic overlap between the optimization variables. Due to this, classical global optimization algorithms (particle swarm, genetic evolution, simulated annealing, etc.) cannot be employed for an efficient solution of this problem. Therefore, we adopted a simpler random search (RS) strategy [40] followed by local refinement (LR). The RS stage generates a random set of uniformly distributed solutions within the optimization boundaries. This step also has to make sure that the generated points fulfill the non-intersection criterion discussed above. The number of initial solutions has to be enough to reasonably cover all qualitatively different array arrangements across the bolus surface. Intuitively, it can be expected that the translation of one element of the array in any direction by an amount smaller than λ/2 does not result in a qualitatively different illumination of the body. This is also the rationale behind the choice of number of grid points in Section 2.5. At the same time, increasing the number of array elements ($n_{c}$) produces more redundancy among a set of solutions, because different antennas can end up covering the same spot. Consequently, we estimated the number of initial random solutions to be generated as:(17)$$n_{r}=\mathrm{round}\left(n_{p}/n_{c}\right)$$
where $n_{p}$ is the number of triangular patches available from the interpolation grid (which is inversely related to the minimum wavelength in water).

Once all $n_{r}$ arrangements have been evaluated, the optimization enters the LR stage, which is implemented using fmincon from MATLAB’s library. This function easily allows for the inclusion of the non-linear constraints, Equation (Equation 14). To reduce the computational time, we sorted the randomly generated solutions in ascending order according to their cost. We then applied the LR to the first solution, obtaining the minimum achievable HCQ for this qualitative arrangement. We proceeded with the next solution until the refined HCQ became worse, thereby assuming that the remaining qualitative arrangements were not likely to yield more favourable SAR patterns. The overall array design procedure is summarized in Figure 10.

Here, we note that the rationale behind developing the analytical expressions reported in Section 2.5 and geometrical expressions for the bolus shape and the antenna coordinate system with respect to the spherical surface coordinates (θ, ϕ), is to make the landscape of the cost-function (HCQ) as smooth as possible with respect to the array optimization variables (θ and ϕ themselves). This is crucial for the LR step, which requires the gradients of the cost function to be numerically evaluated with respect to each optimization variable.

### 2.8. Design Validation

We prepared 8 optimized array designs of increasing order: $n_{c}\in \left[01,02,03,04,06,08,10,12\right]$. For each arrangement, we performed a full array simulation in COMSOL. We compared the predicted and the actual SAR distributions according to the following metrics: (18)$$\begin{array}{cc} \epsilon_{\mathrm{DIS}} & =\int_{M}\frac{|{\mathrm{SAR}}^{\mathrm{SIM}}-{\mathrm{SAR}}^{\mathrm{INT}}|}{|{\mathrm{SAR}}^{\mathrm{SIM}}|}dM\;/\;┌M┘\\ \eta_{H\text{-}S} & =┌H^{\mathrm{SIM}}\cap H^{\mathrm{INT}}┘\;/\;┌H^{\mathrm{SIM}}┘\\ \eta_{C\text{-}S} & =┌C^{\mathrm{SIM}}\cap C^{\mathrm{INT}}┘\;/\;┌C^{\mathrm{SIM}}┘ \end{array}$$
where *H* denotes the hot-spot sub-volume mask (highest *q*-percentile of remaining healthy tissue) and *C* denotes the cold-spot sub-volume mask (lowest *p*-percentile of target volume). While $\epsilon_{\mathrm{DIS}}$ denotes an error metric (the lower the better), $\eta_{H\text{-}S}$ and $\eta_{C\text{-}S}$ are coverage metrics (the higher the better).

To quantify the overall improvement in heating capability of the optimized arrays, we carried out thermal simulations to evaluate the clinically relevant hyperthermia indexes $T_{50}$ and $T_{90}$ for each treatment plan. We also prepared a set of “canonical” applicator designs consisting of one or two rings of equally spaced antennas for $n_{c}\in \left[06,08,10,12\right]$ and report their achieved temperature indexes. These applicators are shown in Figure 11. Since the canonical designs might violate the constraints relative to the avoidance of the eyes, during the evaluation of the treatment plans we turned off the channels relative to the antennas that overlap with the projected eye locations. This resulted in channel 02 being turned off in the canonical applicator design of order $n_{c}=10$, while in the applicator of order $n_{c}=12$ this applies to channels 02 and 03.

## 3. Results

### Grid Simulation

The 221 simulations of the interpolation grid took about 200h on a 32 cores Intel Xeon 2.90 GHz system with 192 Gb of RAM. For comparison, a full eight-channel array simulation takes around 1h on the same computer system, while the interpolated approximation of the same array takes about 15 s, yielding a speedup of roughly 240 times. As the optimizer evaluates around 2000 potential array configurations to determine the best arrangement for eight antennas, the use of the approximation method renders the global optimization feasible. The numbers are even more compelling for higher array sizes.

An example of interpolated versus simulated SAR distribution is shown in Figure 12 for the optimized array design of size $n_{c}=08$ (applicator shown in Figure 13j,n). The two distributions agree well qualitatively. The relative error becomes unacceptable (≫50%) only in regions far from the antennas, such as the mouth (Figure 12e), where the SAR intensity is almost negligible. The cold spot is predicted with high accuracy ($\eta_{C\text{-}S}=81\%$), while the hot-spot identification suffers the most from the approximation error ($\eta_{H\text{-}S}=46\%$).

Figure 14 reports the results of the analysis of the approximation for a single antenna at locations of increasing distance from a simulated grid point. As the selected patch is the largest triangle across the grid, this represents a worst-case scenario. The overall average distribution error $\epsilon_{\mathrm{DIS}}$ reaches a peak of almost 30% when the query location is near the center of the patch. This error is mainly due to a difference in amplitude, as can be seen from Figure 14b. The phase is approximated with the highest accuracy.

The optimized applicator designs for each array size are shown in Figure 13. These should be compared with the location and shape of the target volume, recall Figure 1. The designs of order up to four consistently placed an antenna in the closest proximity to the distal part of the tumor. Beginning from order four, an antenna was also placed on the opposite, frontal side of the head. The design for $n_{c}=6$ closely resembles a canonical one with two interleaved rings of three antennas.

The treatment plans prepared using the optimized designs yielded the values of HCQ and temperature indexes shown in Figure 15. The figure also reports the corresponding values for the canonical designs. The HCQ predicted by the interpolated distribution follows quite closely the actual HCQ evaluated on the simulated distribution, except for the 10-antennas canonical case, which, however, performs poorly in terms of target temperature increase. The relative changes in interpolated 1/HCQ values correlate well with the variations in temperature indexes for both canonical and optimized designs. The only exception is the 12-antennas optimized case, likely due to the main hot spot becoming superficial, as discussed in the following section. The improvement in $T_{50}$ from the best canonical solution ($n_{c}=8$) to the best optimized solution ($n_{c}=10$) is ≈$0.2{\;}^{{}^{\circ}}$C. The improvement in $T_{90}$ from the best canonical solution ($n_{c}=8$) to the best optimized solution ($n_{c}=10$) is ≈$0.3{\;}^{{}^{\circ}}$C.

The SAR and temperature distributions relative to the plans obtained with each optimized design are reported in Figure 16 and Figure 17. The progressive inclusion of more antennas reduces the cranial–caudal elongation of the hot-spot volumes in SAR and simultaneously shifts them closer to and more uniformly surrounding the target volume, which is the desired behavior. The hot-spot masks in SAR follow well the actual resulting location of the temperature peak, except for the 12-antennas case. Here, the hot spot becomes superficial and the SAR prediction degrades. In the majority of dense-array applicator designs, however, the limiting hot spot arises in the pocket of CSF caudal to the target volume.

Table 1 reports the average relative approximation errors and spot mask coverage of each SAR distribution, for both optimized and canonical designs. It is interesting to notice that the distribution error and the mask coverage do not necessarily agree. In particular, $\epsilon_{\mathrm{DIS}}$ is relatively high for the smaller array sizes 02 and 03, but the spot identification has a high degree of accuracy ($\eta_{H\text{-}S},\eta_{C\text{-}S}>79\%$). On the contrary, $\epsilon_{\mathrm{DIS}}$ diminishes for denser arrays, but the hot-spot identification $\eta_{H\text{-}S}$ becomes worse. While the distribution errors reach, at most, ≈30% in the majority of cases, the canonical design of size 10 experiences a remarkably larger approximation error of 58%. This also reflects in a poor hot-spot identification score of only 41%.

To further investigate the reasons behind the failed approximation of the regular ring of 10 antennas, we show the simulated and approximated SAR distributions of this array in Figure 18. The distributions suggest that the inaccuracy stems from a misidentification of the hot spot, which is, in turn, due to a large relative error in the SAR values in the regions of high intensity located cranial and caudal to the target. To understand where this error originates, we show the array arrangement in Figure 19 and the steering power of each channel from the HCQ-optimal treatment plan in Table 2. We further report the relative distribution, amplitude, phase and direction errors of the individual antenna fields in Table 3. Note that the approximations are now severely affected by the amplitude error introduced by the coupling effects between antennas in the array, which is expected. This error is often above 100% because of the high relative errors in the regions of the model subjected to low field intensity. Nevertheless, antennas 02, 06, 07 and 09 exhibit a substantial error above average in the amplitude approximation at the 250MHz operating frequency, which is the main contributor to the total treatment power. Antenna 02 can be disregarded, since its power is zero in the treatment plan. Antenna 07 is also not likely to play a role in the highlighted hot-spot areas, as these lie far from the antenna location. Antennas 06 and 09, on the other hand, are closer to the target and illuminate it from opposite sides with relatively high power. Their deposition patterns interfere precisely in the hot-spot regions and produce an amplitude error that results in an inaccurate prediction of the HCQ value.

## 4. Discussion

The purpose of the approximation method developed in the present work is to facilitate the qualitative evaluation of a large number of array configurations prior to the HT treatment of a brain cancer patient. The assessment of the treatment plans was performed in terms of clinically established parameters, i.e., median temperature $T_{50}$ and 90-percentile temperature $T_{90}$ [6,41,42]. Due to the added computational complexity of thermal simulations, however, the direct assessment of the temperature distribution for thousands of array configurations becomes impractical. The proposed SAR-based field approximation method circumvents this limitation and enables the qualitative evaluation of a given antenna arrangement within seconds. Together with our previously devised SAR-based iterative time-reversal multi-frequency treatment-plan optimization [38], these tools can be used in combination with optimization algorithms to refine an applicator design for a specific patient.

We argue that the proposed approximation method is accurate enough for relative comparison between different design solutions. To support this conclusion, we needed to address the two specific cases (canonical $n_{c}=10$ and optimized $n_{c}=12$) where the approximation method performed worst. In the first case (canonical $n_{c}=10$), the error arose already in the SAR distribution, and the reason for this is the imprecise prediction of the field amplitudes in the zones around the target. The error originates in the inaccurately modelled interference of two antenna fields. As the method consists of extensive approximations, the appearance of outliers is to be expected. A possible workaround could be a local refinement of the interpolation grid in the regions where one anticipates strong radiation powers. Nevertheless, even at this rather coarse sampling, the relative predictions are still correct: the approximated trend of HCQ for the canonical solutions follows the simulated one, and these are reflected in corresponding variations in $T_{50}$ and $T_{90}$, which means that the method can be used for qualitative assessment.

However, in the second case (optimized $n_{c}=12$), the relative improvement in HCQ was also correctly predicted, but this was not reflected in a temperature increase. The reason must be traced to the shift in the location of the most prominent hot spot. While such a limiting hot spot is located in the pocket of CSF caudal to the target volume for the treatment plans relative to the optimized dense arrays $n_{c}=\left[06,08,10\right]$, in the optimized $n_{c}=12$ case the peak temperature was reached near the superficial part of the tumor (Figure 17n,s). We have previously shown that 1/HCQ correlates well with the temperature indexes $T_{50}$ and $T_{90}$ for deep targets, but the correlation quickly deteriorates for superficial targets if the water bolus directly affects the temperature distribution in the target volume [28], as the SAR distribution can no longer predict the location and severity of each spot. One can, thus, speculate that an analog mechanism lies behind this result. Improved temperatures might be achieved with more aggressive water-bolus cooling to suppress the superficial hot spot and restore the SAR-temperature correlation. Alternatively, a thinner exclusion layer in the SAR evaluation mask might guide the optimizer towards solutions that deposit less power in the superficial zone. In this study, we applied a 20mm exclusion, which is on the upper limit of typical cooling depths for clinical water boluses employed in superficial hyperthermia treatments [15].

The comparison between optimized and canonical arrays reveals only a moderate gain in indexed temperatures (≈$0.2{\;}^{{}^{\circ}}$C in $T_{50}$ and ≈$0.3{\;}^{{}^{\circ}}$C in $T_{90}$, Figure 15, difference between best canonical and best optimized solution). This is expected, as the present canonical designs are in fact already tailored to the patient in terms of antenna design and bolus shape. Nevertheless, the achieved gains are still clinically relevant since an increase by ≈$0.3{\;}^{{}^{\circ}}$C in $T_{90}$ would correspond to an increase by ≈1.5 in thermal dose when $T_{90}$ is below the breakpoint value of $43{\;}^{{}^{\circ}}$C, according to the CEM43T90 model [42]. This is further supported by the consideration that measured temperature changes have been shown to reflect the relative variations predicted by numerical simulations with a precision as small as $0.1{\;}^{{}^{\circ}}$C [43]. One could expect even larger differences for tumors located higher in the supratentorial region where the traditional conformal ring design is presumably less effective at delivering the dose due to the geometry of the vertex.

The need to down-sample the patient model from 1mm to 4mm, in order to maintain the simulation of the interpolation grid within reasonable time, might represent a limitation in the proposed method. The detail of the segmentation of the CSF, in particular, has been shown to affect the resulting temperature profiles [44]. The CSF can present features below 4mm, especially in the layer between the skull and brain. In this particular medulloblastoma model, however, we can still afford such a coarse meshing as the limiting hot spot consistently arises in the deep pocket of CSF adjacent to the target. We verified that this is the case even when a denser grid of 1mm is utilized for the simulation [45]. The CSF in this region is sufficiently captured by a 4mm resolution for the purpose of assessing qualitative antenna arrangements for this target. However, this may not be the case when other tumor locations and sizes are considered. Glioblastoma patients, for instance, would likely require finer meshing to account for the occurrence of hot spots in the distal layers of CSF.

As a final note, we address the question on whether it is meaningful to consider applicator designs with such a degree of customization for a certain patient, especially when it is already a challenge to accurately model and position patients in much simpler applicator designs [46]. In our opinion, the rationale behind this contribution lies in addressing a particularly challenging anatomical region, the brain, and strive for a design that will eventually enable hyperthermia treatments in this organ. Such treatments might require a higher degree of customization than current clinical solutions. The method allows us to qualitatively sift through many potential array configurations and select the most suitable one for a certain tumor size, shape and location. In a practical setting, we envision this method being used for the development and manufacturing of a limited set of (head) applicators, each with a qualitatively different antenna arrangement and optimized for a specific target location. During the treatment planning stage, the applicator with the best heating capability for the patient at hand can be determined and selected. Unfortunately, a direct comparison with current clinical applicators is not possible, as these were not intended for brain-tumor treatment. Furthermore, any comparison with the absolute temperatures reported in the literature would be affected by the considerable uncertainties in thermal simulations [47] and their strong dependence on the specific patient modeling.

## 5. Conclusions

We proposed and validated, by means of numerical comparisons, a novel field-approximation method for the fast evaluation of different antenna arrangements in a helmet applicator for intracranial microwave hyperthermia treatments. The method was further used in conjunction with a fast multi-frequency treatment-plan optimization scheme to improve the design of an applicator for a specific paediatric brain-cancer patient. The method is accurate enough to provide qualitative indications about the most suitable antenna arrangement for a given tumor shape and location. The technique can be particularly useful in the design of UWB applicators where the classical single-frequency array theory used for narrow-band applicators might prove insufficient to achieve an optimal configuration. Further studies are required to assess the sensitivity of the proposed technique to the resolution of the interpolation grid, and future developments might involve the inclusion of the antenna polarization angles in the set of design parameters.

## Figures and Tables

**Figure 1 cancers-15-01447-f001:**
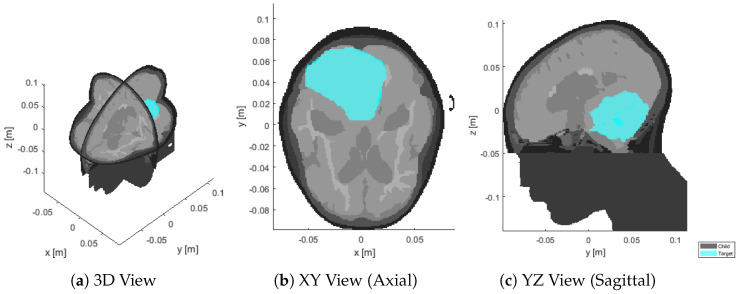
Sections of the segmented patient model (gray) at the original resolution of 1mm, with superimposed target volume (cyan).

**Figure 2 cancers-15-01447-f002:**
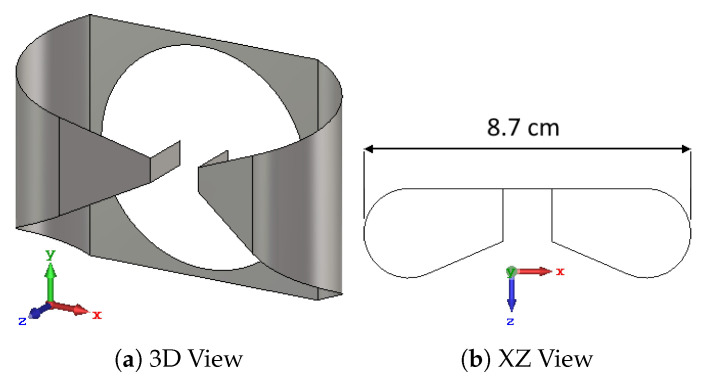
Self-grounded bow-tie antenna optimized for the 250–500 MHz band. The antenna’s polarization axis *u* is aligned with the *x* axis (red), while its main directivity axis *w* is aligned with the *z* axis (blue). The center of the antenna’s local coordinate system corresponds to the center of its ground plate, which is also the center of the circular feed opening. The overall dimensions are 8.7cm along *x*, 6.2cm along *y*, and 2.4cm along *z*.

**Figure 3 cancers-15-01447-f003:**
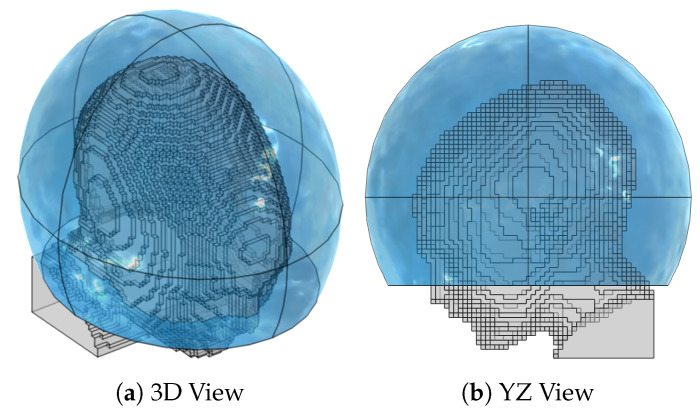
Patient model (gray) down-sampled to a 4mm resolution, together with the water bolus shape (blue). The ellipsoid is designed to maintain a bolus thickness as close as possible to 5cm around the scalp and is clipped right above the shoulders and nostrils to allow for breathing. The resulting bolus dimensions are 25.0cm along the left-right axis, 28.4cm along the anterior-posterior axis, and 22.1cm along the cranial-caudal axis.

**Figure 4 cancers-15-01447-f004:**
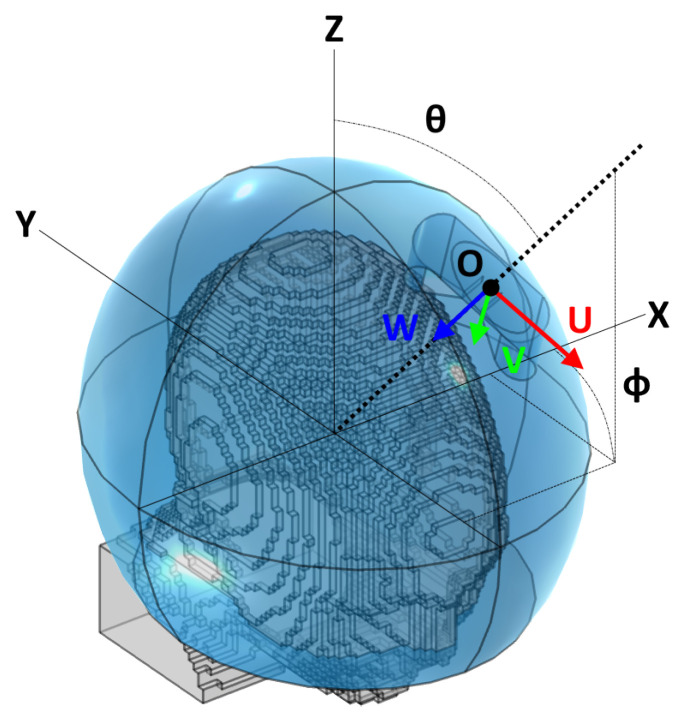
Reference schematic for the arrangement of a single antenna. Note that the angle ϕ, while following the classic right-hand convention, is shown here on the negative *y* half-space for readability. The figure refers to a local coordinate system centered at the ellipsoid’s center and aligned with the global cartesian axes.

**Figure 5 cancers-15-01447-f005:**
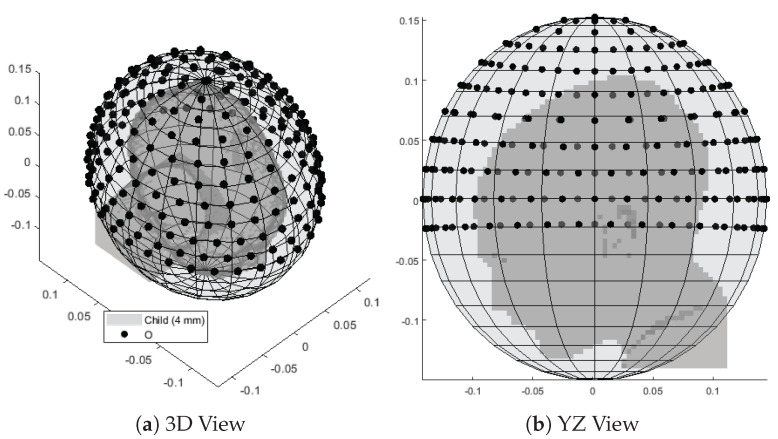
Interpolation grid made of 221 points (black) uniformly distributed around the child patient model (gray) and lying on the surface of a fitted ellipsoid. The average distance between pairs of nearby points is 2.6cm.

**Figure 6 cancers-15-01447-f006:**
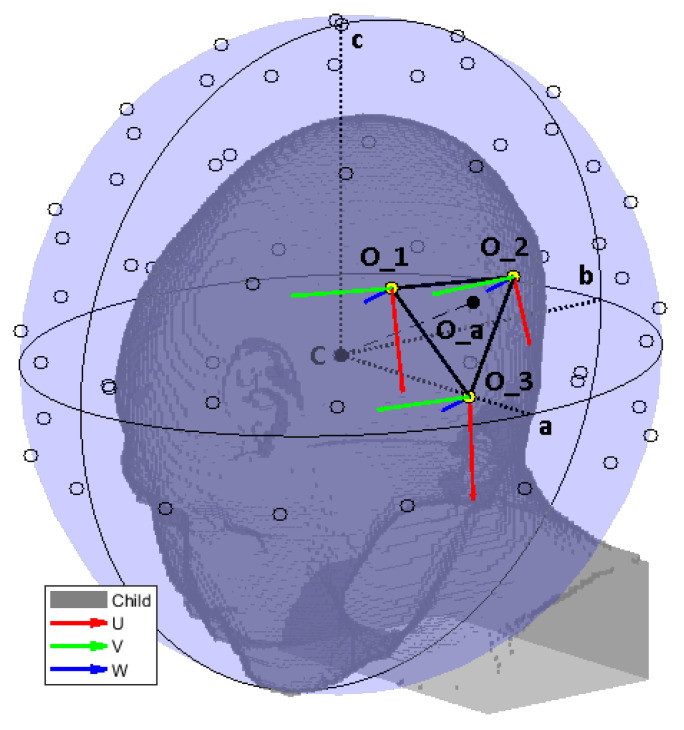
Reference schematic for the field interpolation procedure, using a less dense grid to facilitate reading. The ellipsoid is shown in its entirety to highlight the different radii. However, in the actual simulation model, the bolus was clipped at the level of the shoulders. We show the ellipsoid center *C* and its radii *a*, *b*, *c*. The interpolation grid is shown with black circles. The selected interpolation patch $\left(O_{1},O_{2},O_{3}\right)$ for an antenna at location $O_{a}$ is highlighted with thick black edges and yellow vertices. The local coordinate systems of the selected grid points are also shown. An equivalent system was built for the query antenna location $O_{a}$.

**Figure 7 cancers-15-01447-f007:**
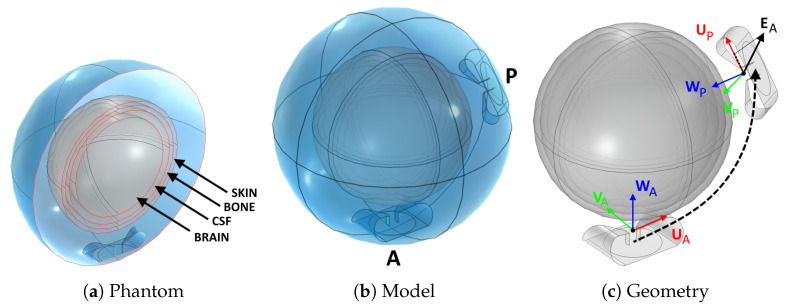
Procedure to determine the coupling between antenna pairs. A spherical brain phantom (**a**) was inserted into a spherical bolus (**b**). An active (A) and a passive (P) antenna were added inside the bolus. First, the individual fields $E_{A}$ and $E_{P}$ of each antenna were determined without the presence of the other antenna. Subsequently, the active antenna was excited with the presence of the passive antenna and the overall coupled field $E_{A+P}$ was determined. A correlation factor between the coupled field $E_{A+P}$ and the passive antenna field $E_{P}$ was determined. This was found to be proportional to the projection on $U_{P}$ of the individual field $E_{A}$ at the location of the passive antenna (**c**).

**Figure 8 cancers-15-01447-f008:**
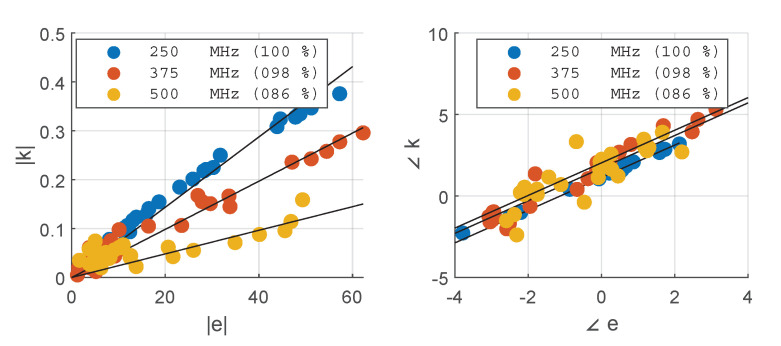
Correlation between the projection $e_{AP}$ of the active antenna’s field $E_{A}$ on the passive antenna’s polarization axis $U_{P}$ at $O_{P}^{\prime }$, and the coupling coefficient $k_{AP}$ obtained by decorrelation of the remainder field $E_{A+P}-E_{A}$ with respect to $E_{P}$. The results are reported for each frequency in the operating set. The solid black lines show the fitted complex coupling coefficient *c*, while the legends report the correlation coefficients for each fit. The fit was carried out on the complex values.

**Figure 9 cancers-15-01447-f009:**
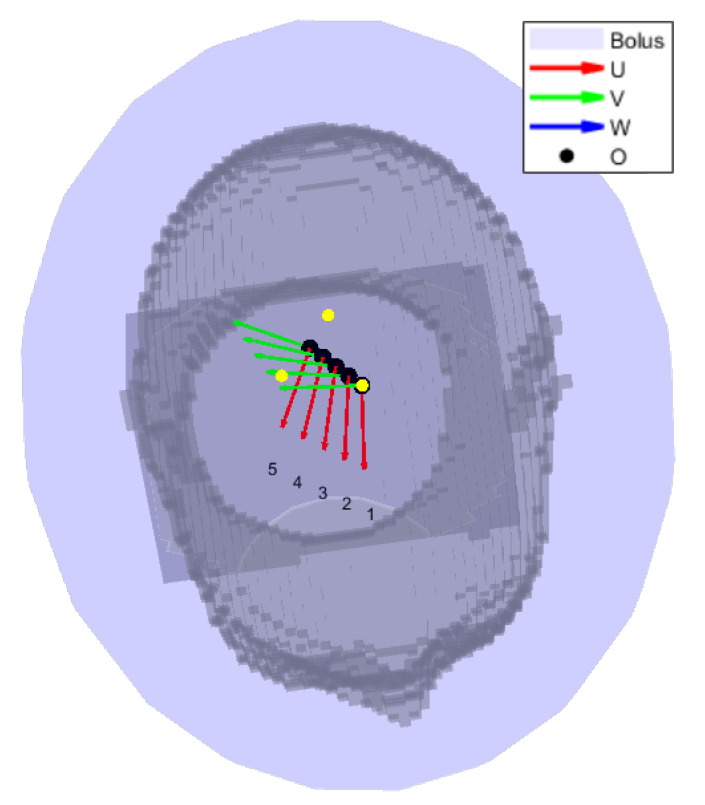
Location sweep for the sensitivity analysis of the field interpolation error. The black circles represent the interpolation grid points. The yellow dots are the grid points selected for interpolation, and are the corners of the triangular patch of largest area. The gray shade is the patient in bird’s eye view. The local coordinate systems of each antenna location to be approximated are shown as superimposed triplets.

**Figure 10 cancers-15-01447-f010:**
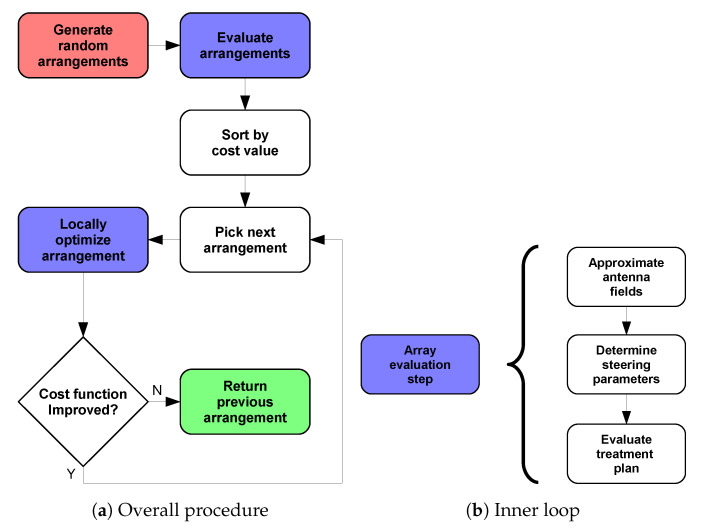
Applicator optimization procedure to determine the best antenna arrangement for a given patient. The procedure begins at the red step and ends at the green step. The steps highlighted in blue involve the sub-steps shown in (**b**) to determine the cost-function value of a certain array arrangement.

**Figure 11 cancers-15-01447-f011:**
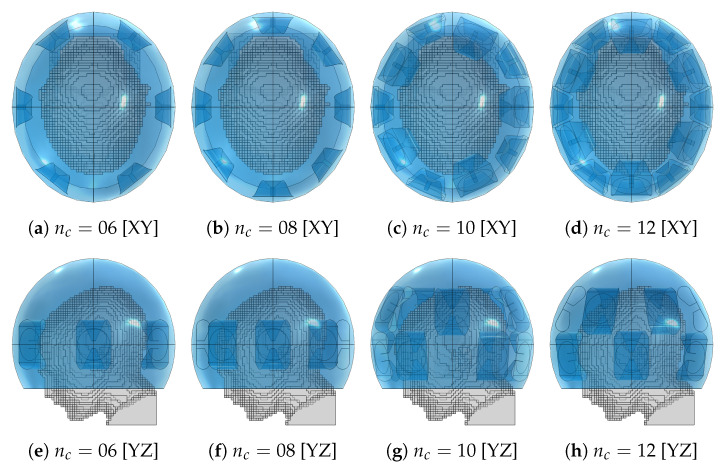
Canonical applicator designs with increasing number of antenna elements ($n_{c}$) for the medulloblastoma pediatric patient model (gray) using the fitted ellipsoidal water bolus shape (blue).

**Figure 12 cancers-15-01447-f012:**
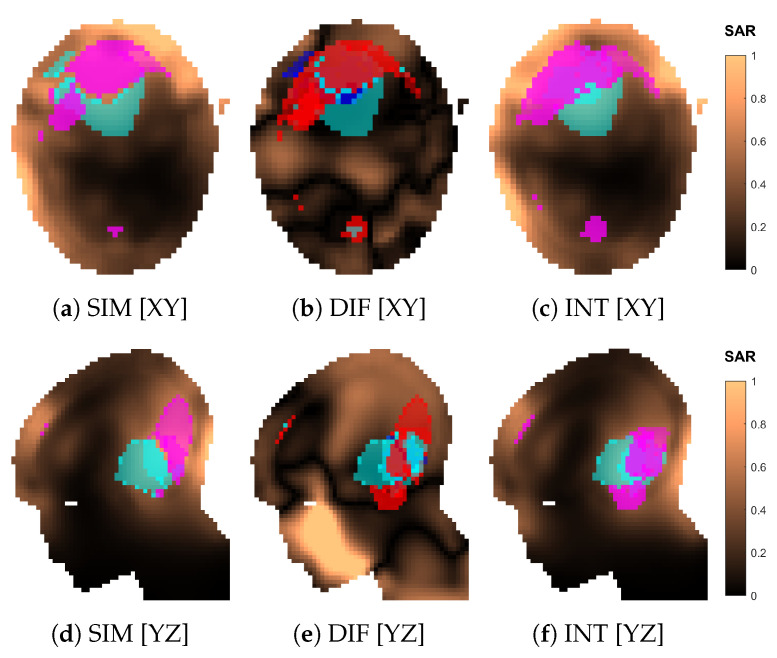
Comparison of the normalized SAR distributions obtained via approximation (INT) and full simulation (SIM) for the optimized applicator design of order $n_{c}=08$. Sections of the SAR distribution inside the patient model, taken at the target center. The volumes in magenta represent the highest *q*-percentile in the remaining healthy tissue (hot spot), while the volumes in cyan represent the lowest *p*-percentile in the target (cold spot). The difference (DIF) distribution is relative to the simulated one, i.e., ${\mathrm{SAR}}^{\mathrm{DIF}}=|{\mathrm{SAR}}^{\mathrm{SIM}}-{\mathrm{SAR}}^{\mathrm{INT}}\left|/\right|{\mathrm{SAR}}^{\mathrm{SIM}}|$. In (**b**,**e**), the volumes in magenta represent hot-spot coverage ($H^{\mathrm{SIM}}\cap H^{\mathrm{INT}}$), while the volumes in cyan represent cold-spot coverage ($C^{\mathrm{SIM}}\cap C^{\mathrm{INT}}$). The volumes in red represent hot-spot exclusion ($H^{\mathrm{SIM}}⊕H^{\mathrm{INT}}$), while the volumes in blue represent cold-spot exclusion ($C^{\mathrm{SIM}}⊕C^{\mathrm{INT}}$).

**Figure 13 cancers-15-01447-f013:**
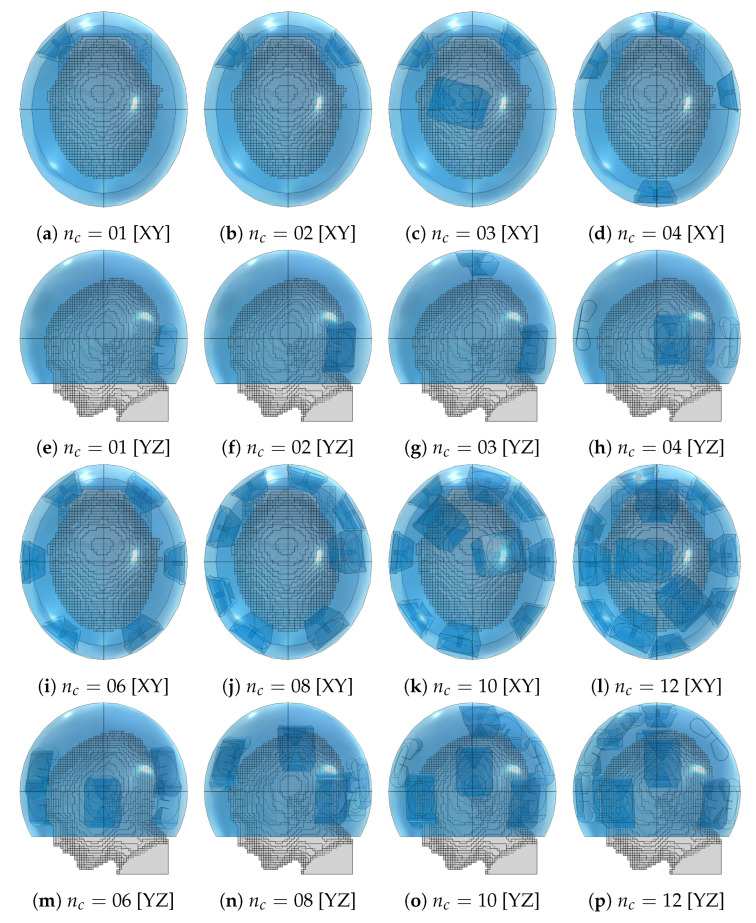
Optimized applicator designs with increasing number of antenna elements ($n_{c}$) for the medulloblastoma pediatric patient model (gray) using the fitted ellipsoidal water bolus shape (blue).

**Figure 14 cancers-15-01447-f014:**
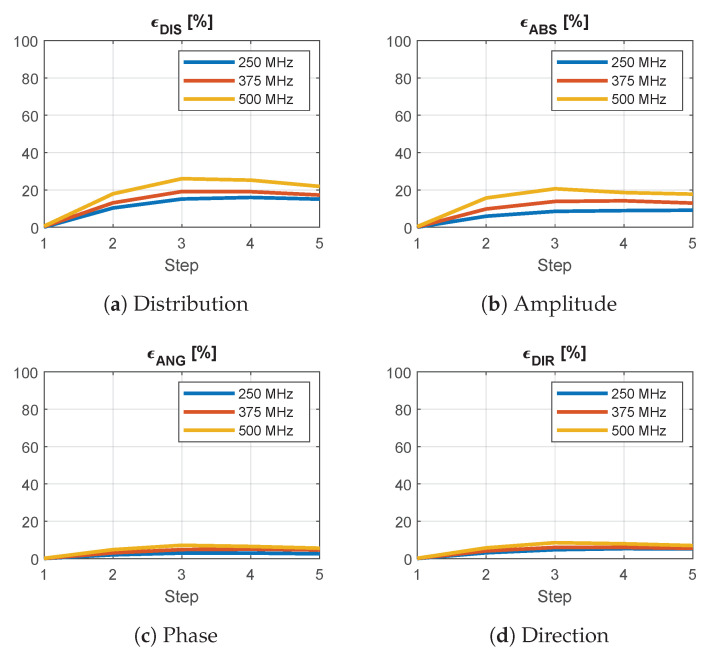
Average relative error between the interpolated and simulated E-field distributions of a single antenna at increasing distance from a grid point for different frequencies across the operating band. The step indicates the position of the antenna within the interpolation patch, where zero corresponds to one of the simulated corners. A phase error $\epsilon_{\mathrm{ANG}}$ of 100% means that the fields are in opposition. A direction error $\epsilon_{\mathrm{DIR}}$ of 100% means that the fields are orthogonal.

**Figure 15 cancers-15-01447-f015:**
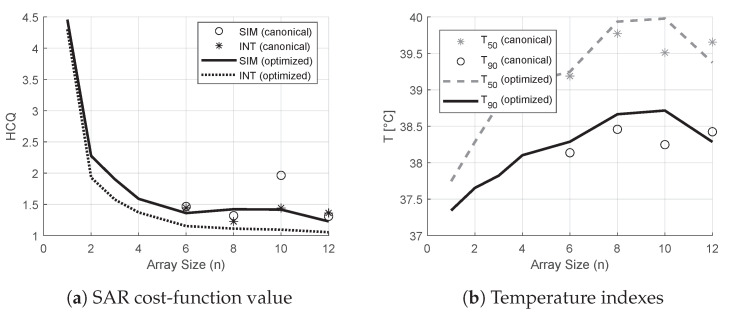
Values of HCQ, $T_{50}$ and $T_{90}$ relative to the treatment plans prepared using canonical and optimized applicator designs of increasing order (line plots). The values for the canonical applicator designs are also reported as scatter plots. In SAR, the value of HCQ predicted by the field approximation is compared against the value from the actual simulated field.

**Figure 16 cancers-15-01447-f016:**
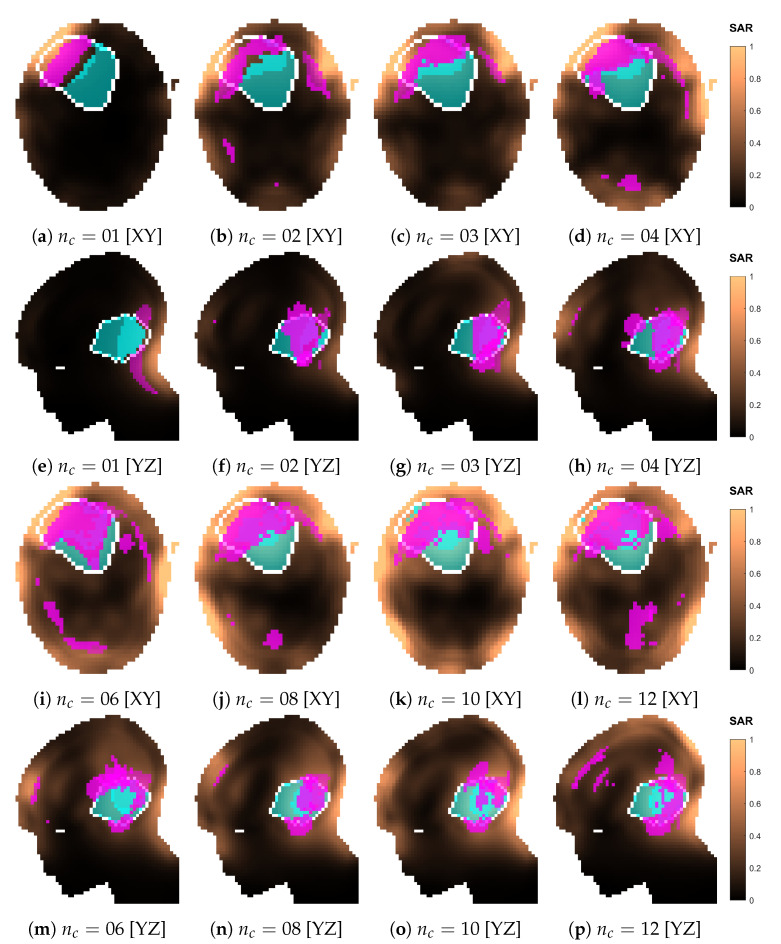
Normalized SAR distributions relative to each optimized applicator design with increasing number of antenna elements ($n_{c}$). Sections taken at target center. The white line delineates the target volume. The volumes in magenta represent the highest *q*-percentile (2.8%) in the remaining healthy tissue (hot spot), while the volumes in cyan represent the lowest *p*-percentile (50%) in the target (cold spot).

**Figure 17 cancers-15-01447-f017:**
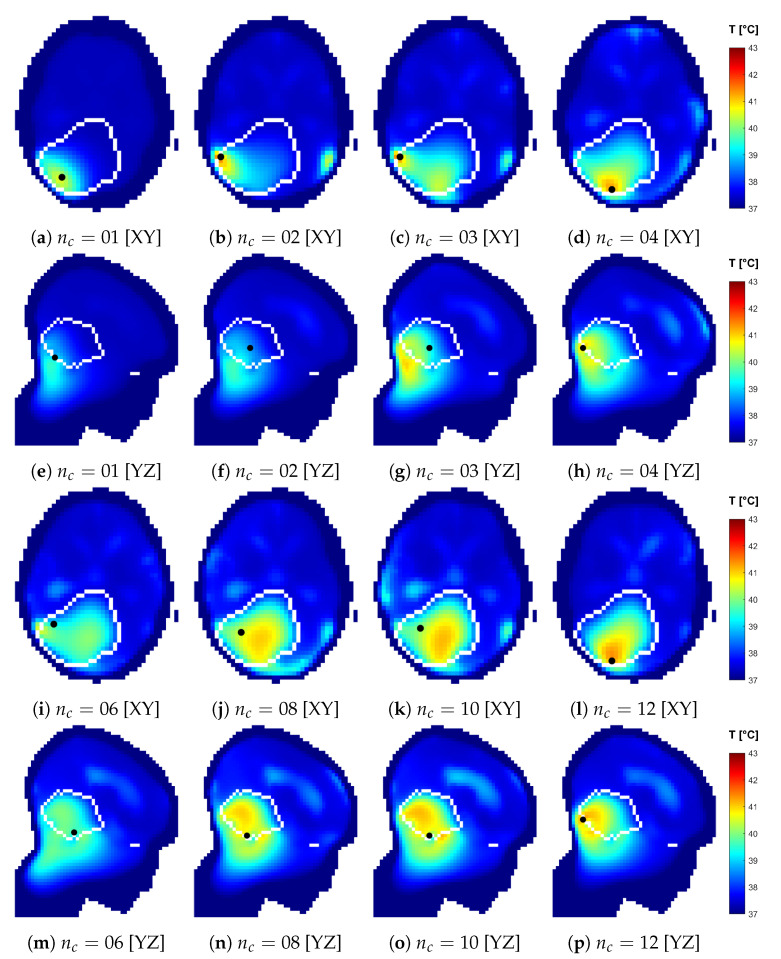
Temperature distributions relative to each optimized applicator design with increasing number of antenna elements ($n_{c}$). Sections taken at target center. The views are flipped to show the side where the temperature peak in the healthy tissue is located, marked with a black dot, which is located off plane with respect to the sections. The white line delineates the target volume.

**Figure 18 cancers-15-01447-f018:**
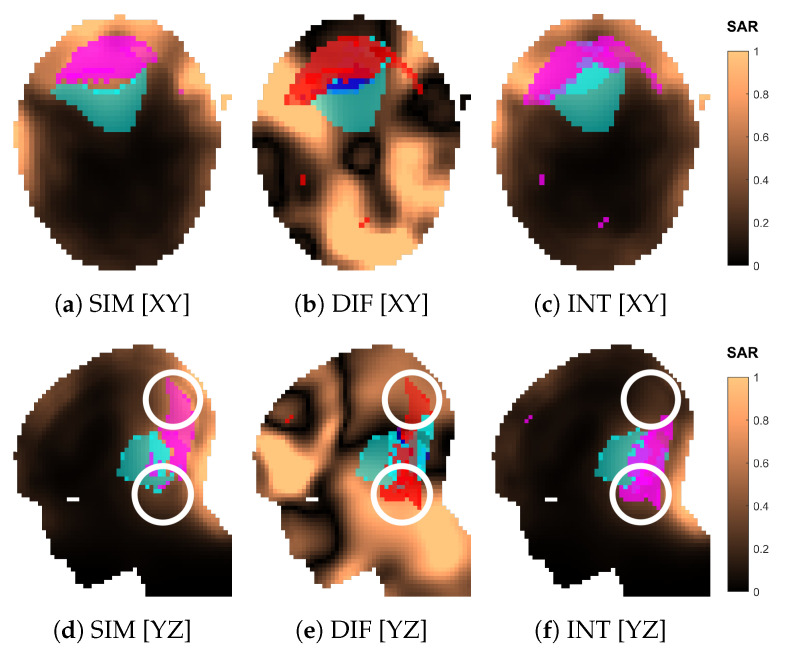
Comparison of the normalized SAR distributions obtained via approximation (INT) and full simulation (SIM) for the canonical applicator design of order $n_{c}=10$. Sections of the SAR distribution inside the patient model, taken at the target center. The volumes in magenta represent the highest *q*-percentile in the remaining healthy tissue (hot spot), while the volumes in cyan represent the lowest *p*-percentile in the target (cold spot). The difference (DIF) distribution is relative to the simulated one, i.e., ${\mathrm{SAR}}^{\mathrm{DIF}}=|{\mathrm{SAR}}^{\mathrm{SIM}}-{\mathrm{SAR}}^{\mathrm{INT}}\left|/\right|{\mathrm{SAR}}^{\mathrm{SIM}}|$. In (**b**,**e**), the volumes in magenta represent hot-spot coverage ($H^{\mathrm{SIM}}\cap H^{\mathrm{INT}}$), while the volumes in cyan represent cold-spot coverage ($C^{\mathrm{SIM}}\cap C^{\mathrm{INT}}$). The volumes in red represent hot-spot exclusion ($H^{\mathrm{SIM}}⊕H^{\mathrm{INT}}$), while the volumes in blue represent cold-spot exclusion ($C^{\mathrm{SIM}}⊕C^{\mathrm{INT}}$). The white circle in (**d**–**f**) highlights the location of the hot-spot misidentification.

**Figure 19 cancers-15-01447-f019:**
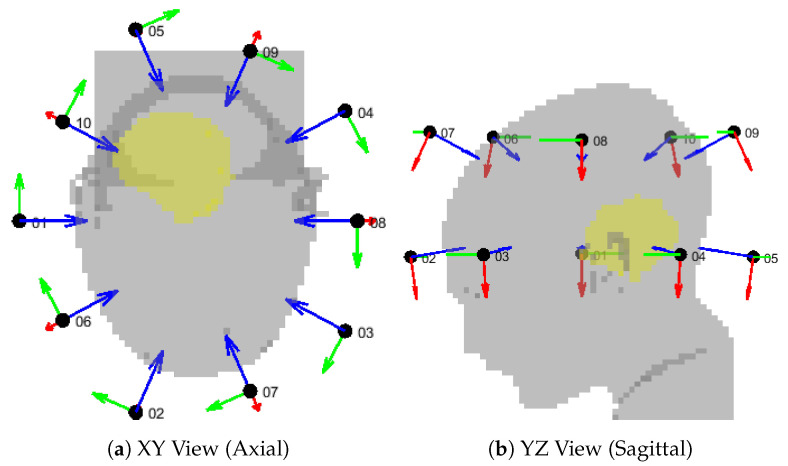
Geometrical setup for the canonical applicator design of order $n_{c}=10$. The patient model (gray) is shown together with the antenna local coordinate systems, where the red vector is *U*, the green vector is *V*, and the blue vector is *W*. The antenna center is represented by a black dot and labeled with the channel number. The target volume is highlighted in yellow.

**Table 1 cancers-15-01447-t001:** Error indicators of the overall approximated SAR distributions with respect to the corresponding distributions obtained from full simulations. $\epsilon_{\mathrm{DIS}}$ distribution error, $\eta_{H\text{-}S}$ hot-spot mask error, $\eta_{C\text{-}S}$ cold-spot mask error. Both optimized (OPT) and canonical (CAN) array designs are reported. The last four rows report the same error indicators for the densest optimized array and increasing propagation order *K* for the coupling modeling.

	$\epsilon_{\mathrm{DIS}}\;\left[\%\right]$	$\eta_{H\text{-}S}\;\left[\%\right]$	$\eta_{C\text{-}S}\;\left[\%\right]$
$n_{c}=01$ (OPT)	08	98	99
$n_{c}=02$ (OPT)	32	80	95
$n_{c}=03$ (OPT)	36	79	96
$n_{c}=04$ (OPT)	22	63	87
$n_{c}=06$ (OPT)	28	54	84
$n_{c}=08$ (OPT)	29	46	81
$n_{c}=10$ (OPT)	32	46	82
$n_{c}=12$ (OPT)	28	65	84
$n_{c}=06$ (CAN)	26	46	88
$n_{c}=08$ (CAN)	20	69	91
$n_{c}=10$ (CAN)	58	41	84
$n_{c}=12$ (CAN)	25	57	85
$n_{c}=12$ (OPT) [K=1]	42	62	86
$n_{c}=12$ (OPT) [K=2]	34	71	94
$n_{c}=12$ (OPT) [K=3]	28	65	84
$n_{c}=12$ (OPT) [K=4]	29	72	95
$n_{c}=12$ (OPT) [K=5]	28	44	78

**Table 2 cancers-15-01447-t002:** Normalized power radiated by each antenna according to the channel steering parameters of the HCQ-optimal solution for the 10-elements canonical array design. The last row and column report the total power per frequency and per antenna, respectively.

POWER [%]	250 MHz	375 MHz	500 MHz	Antenna Total:
Ant. 01	06	04	05	15
Ant. 02	00	00	00	00
Ant. 03	02	01	03	05
Ant. 04	07	08	02	17
Ant. 05	16	05	02	23
Ant. 06	03	01	02	06
Ant. 07	01	06	03	10
Ant. 08	01	03	04	09
Ant. 09	05	01	04	09
Ant. 10	04	01	02	07
Frequency total:	44	30	26	

**Table 3 cancers-15-01447-t003:** Error indicators of the approximated E-field distributions of each individual antenna in the 10-elements canonical array design, with respect to the corresponding distributions obtained from the full-array simulation. $\epsilon_{\mathrm{DIS}}$ distribution error, $\epsilon_{\mathrm{ABS}}$ amplitude error, $\epsilon_{\mathrm{ANG}}$ phase error, $\epsilon_{\mathrm{DIR}}$ direction error. The last row reports the average error among the set of antennas.

$\epsilon_{\mathrm{DIS}}\;\left[\%\right]$	**250 MHz**	**375 MHz**	**500 MHz**	$\epsilon_{\mathrm{ABS}}\;\left[\%\right]$	**250 MHz**	**375 MHz**	**500 MHz**
Ant. 01	144	099	080	Ant. 01	109	071	046
Ant. 02	201	134	106	Ant. 02	171	117	072
Ant. 03	153	111	091	Ant. 03	127	094	069
Ant. 04	152	106	080	Ant. 04	121	071	051
Ant. 05	150	113	074	Ant. 05	091	095	062
Ant. 06	179	155	096	Ant. 06	151	119	088
Ant. 07	221	161	101	Ant. 07	185	114	102
Ant. 08	155	157	088	Ant. 08	104	105	063
Ant. 09	156	150	096	Ant. 09	160	111	095
Ant. 10	151	144	086	Ant. 10	116	101	055
MEAN:	166	133	090	MEAN:	133	100	070
** $\epsilon_{\mathrm{ANG}}\;\left[\%\right]$ **	**250 MHz**	**375 MHz**	**500 MHz**	** $\epsilon_{\mathrm{DIR}}\;\left[\%\right]$ **	**250 MHz**	**375 MHz**	**500 MHz**
Ant. 01	026	022	018	Ant. 01	023	022	022
Ant. 02	040	035	026	Ant. 02	024	027	024
Ant. 03	032	026	025	Ant. 03	024	025	024
Ant. 04	027	022	020	Ant. 04	023	025	021
Ant. 05	028	023	018	Ant. 05	024	024	021
Ant. 06	028	030	032	Ant. 06	023	024	025
Ant. 07	032	035	032	Ant. 07	022	025	025
Ant. 08	026	031	027	Ant. 08	022	023	021
Ant. 09	030	033	025	Ant. 09	021	024	023
Ant. 10	025	029	025	Ant. 10	021	024	023
MEAN:	029	029	025	MEAN:	023	025	023

## Data Availability

Data available on request due to restrictions, e.g., privacy or ethical. The data presented in this study are stored on secure servers and available on request from the corresponding author.

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
