# Peer review of "Antenna Arrangement in UWB Helmet Brain Applicators for Deep Microwave Hyperthermia"

_cancers, 2023, doi:10.3390/cancers15051447_

Round 1

Reviewer 1 Report

This paper presents an optimization scheme for antenna arrangement to achieve optimal heating on brain tumors in children. The authors focused on techniques that sacrificed spatial resolution over simulation time, which is acceptable at this phase. They have also only tested the presented methods in only one case, which is a weakness. How strong of a weakness, I will leave that to the editor. Given that the novelty is in the optimization scheme, I would classify this weakness as minor. The methods are robust and elegantly developed. I was expecting more impactful results, but the improvement of 0.3°C is not as relevant as presented by the authors. The spatial resolution of 4mm is quite coarse but acceptable, given the high number of simulations performed. There are regions of the CSF near the surface that are thinner than 4mm, so it is unclear how the model was generated at the interface between skull and brain where CSF is thinner than 4mm. Please explain.

Please view the attached and address all 56 questions/comments. I am OK with answers in the document for a more efficient review process. Here are some highlights:

-          Overall, the manuscript is very well written. However, there are some instances of lazy and verbal writing as well as some text that require further clarification.

-          The criteria for choosing the water bolus thickness is not clear.

-          Some mathematical errors require revision, such as using T for target and temperature.

-          Images and tables should stand by themselves, meaning the reader shouldn't be forced to search the text to understand the image. The label font size needs to be increased, dimensions need to be added, and more description in the caption is required. Parameters listed in the images need to be introduced in the caption. And please add T (°C) and SAR above the scale.

-          The explanation for the two solutions that failed the predicted temperature increase is inadequate. The authors mentioned that a detailed analysis of the individual antenna fields is necessary to unveil the cause behind the outlier. It is not clear why this analysis was not carried out.

-          I would add a few sentences on how you envision applying this concept in practice

 Overall, I consider the results relatively short but sufficient given that the paper is more focused on developing an original method, which can greatly impact future microwave applicators.

Author Response

REVIEWER 1

This paper presents an optimization scheme for antenna arrangement to achieve optimal heating on brain tumors in children. The authors focused on techniques that sacrificed spatial resolution over simulation time, which is acceptable at this phase. They have also only tested the presented methods in only one case, which is a weakness. How strong of a weakness, I will leave that to the editor. Given that the novelty is in the optimization scheme, I would classify this weakness as minor. The methods are robust and elegantly developed. I was expecting more impactful results, but the improvement of 0.3°C is not as relevant as presented by the authors. The spatial resolution of 4mm is quite coarse but acceptable, given the high number of simulations performed. There are regions of the CSF near the surface that are thinner than 4mm, so it is unclear how the model was generated at the interface between skull and brain where CSF is thinner than 4mm. Please explain.

Thank you for this encouraging assessment and thorough analysis.

Indeed, there are regions where the CSF layer can become thinner than 4mm. These are mostly located between the skull and the brain, although some can occur also between the brain hemispheres. In this particular medulloblastoma model, however, we can afford to perform such an aggressive down-sampling because, even at higher resolutions, the limiting hot-spot consistently arises in the deep pocket of CSF adjacent to the large target (and therefore most exposed to SAR deposition). This pocket is still sufficiently captured by the 4mm grid, while we have seen that the deposition in the distal layers is counteracted by bolus cooling (even with a denser grid). This might no longer apply in other patient models with different tumor locations or sizes. We added a paragraph concerning this issue in the discussion.

Please view the attached and address all 56 questions/comments. I am OK with answers in the document for a more efficient review process. Here are some highlights:

We have addressed all the comments in the PDF.

-          Overall, the manuscript is very well written. However, there are some instances of lazy and verbal writing as well as some text that require further clarification.

We addressed the issues where pointed out in the PDF.

-          The criteria for choosing the water bolus thickness is not clear.

The thickness is a compromise between sensitivity of the antenna to anatomical variations and power losses in the bolus. We added this explanation under Section 2.2.

-          Some mathematical errors require revision, such as using T for target and temperature.

We changed the math font for all volume variables (T, R, M).

-          Images and tables should stand by themselves, meaning the reader shouldn't be forced to search the text to understand the image. The label font size needs to be increased, dimensions need to be added, and more description in the caption is required. Parameters listed in the images need to be introduced in the caption. And please add T (°C) and SAR above the scale.

We revised most of the captions, increased the sizes, added dimensions where relevant and included units in the legends.

-          The explanation for the two solutions that failed the predicted temperature increase is inadequate. The authors mentioned that a detailed analysis of the individual antenna fields is necessary to unveil the cause behind the outlier. It is not clear why this analysis was not carried out.

We have now added a thorough analysis of the individual antenna field errors relative to the canonical solution with 10 elements to the results, and a corresponding piece in the discussion.

-          I would add a few sentences on how you envision applying this concept in practice

We believe the method can be used to prepare a limited set of (head) applicators, each optimized for a qualitatively different target (distal, central, etc.). The applicator to be used for a specific patient can then be chosen by a quantitative analysis during the treatment planning process. We added a few sentences about this at the end of the discussion section.

Overall, I consider the results relatively short but sufficient given that the paper is more focused on developing an original method, which can greatly impact future microwave applicators.

Reviewer 2 Report

The manuscript deals with the design of an array of microwave hyperthermia antennas customized for a specific patient. Antennas are arranged in a helmet to deliver hyperthermia to the brain of a pediatric patient. A specific interpolation scheme has been proposed and validated to keep the time required for optimizing antenna positions at an acceptable value. Authors have demonstrated that the proposed design technique in optimizing a helmet applicator for treating a medulloblastoma in a pediatric patient achieves a significant T90 figure-of-merit increase with respect to a conventional ring applicator with the same number of elements.

The manuscript is clear and presented in a well-structured manner. It is scientifically sound. Some numerical experiments are correctly designed to test the hypothesis.

However, minor revisions are required to help the reader and remove minor inaccuracies. Here is a list of items to review. They are listed below in the order they appear on the manuscript.

1.       Page 3, line 96. The term “insertion loss above 10 dB…”, referring to the antenna, should be changed to “return loss above 10 dB…”.

2.       Page3, line 100. Water-bolus external shape is referred to as a “spheroid”, but having three different semi-axes should be referenced throughout the text as an “ellipsoid”.

3.       Page 5, line 188. Section 2.5.1 describe how to place np points on the bolus surface. The reviewer doesn’t understand how minimizing the sum of squared distances between each pair of points (without any other constraint) leads to a sort of uniform distribution of such points. According to the reviewer’s understanding, such a minimization should collapse all points to a single point. Aren’t there other constraints? For example, are some points fixed to be on the border of the bolus?

4.       Page 6, caption of Fig. 4. Instead of “negative Y quadrant” it should be more appropriate “negative Y halfspace”.

5.       Page 7, line 221. At the beginning of the line I suppose that “H^1” must be replaced by “H1”.

6.       Page 8, line 239. The thickness of 6.3 mm for the skin looks strange. Is it correct?

7.       Page 8, line 252. Please specify in which way the passive antenna is terminated at its input port. Open circuit, short circuit, matched load, other?

8.       Page 9, Eq. (12). The exponent of the n x n matrix is (k-1) or k ? I assume that k is the number of reflections that should be accounted for.

NNo further observations.

Author Response

The manuscript deals with the design of an array of microwave hyperthermia antennas customized for a specific patient. Antennas are arranged in a helmet to deliver hyperthermia to the brain of a pediatric patient. A specific interpolation scheme has been proposed and validated to keep the time required for optimizing antenna positions at an acceptable value. Authors have demonstrated that the proposed design technique in optimizing a helmet applicator for treating a medulloblastoma in a pediatric patient achieves a significant T90 figure-of-merit increase with respect to a conventional ring applicator with the same number of elements.

The manuscript is clear and presented in a well-structured manner. It is scientifically sound. Some numerical experiments are correctly designed to test the hypothesis.

We thank the reviewer for the encouraging feedback and attentive comments!

However, minor revisions are required to help the reader and remove minor inaccuracies. Here is a list of items to review. They are listed below in the order they appear on the manuscript.

  1. Page 3, line 96. The term “insertion loss above 10 dB…”, referring to the antenna, should be changed to “return loss above 10 dB…”.

We changed to return loss.

  1. Page3, line 100. Water-bolus external shape is referred to as a “spheroid”, but having three different semi-axes should be referenced throughout the text as an “ellipsoid”.

We changed everywhere to “ellipsoid”.

  1. Page 5, line 188. Section 2.5.1 describe how to place np points on the bolus surface. The reviewer doesn’t understand how minimizing the sum of squared distances between each pair of points (without any other constraint) leads to a sort of uniform distribution of such points. According to the reviewer’s understanding, such a minimization should collapse all points to a single point. Aren’t there other constraints? For example, are some points fixed to be on the border of the bolus?

Correct. We forgot to specify “inverse of the squared distances”. We added this in the text. We also added a sentence concerning the constraints.

  1. Page 6, caption of Fig. 4. Instead of “negative Y quadrant” it should be more appropriate “negative Y halfspace”.

We changed to “half-space”.

  1. Page 7, line 221. At the beginning of the line I suppose that “H^1” must be replaced by “H1”.

Correct, we removed the hat. Thank you for spotting the mistake.

  1. Page 8, line 239. The thickness of 6.3 mm for the skin looks strange. Is it correct?

We agree that the thickness is exaggerate with respect to anatomical knowledge. The reason is the manual segmentation performed by a trained clinician to obtain this model. The number of identified tissues was kept low to reduce the complexity of the task. The “skin” tissue, in this case, includes skin (ε ≈ 4, σ ≈ 0.0002 S/m, about 2.3 mm thickness in adult males), fat (ε ≈ 2.5, σ ≈ 0.035 S/m, about 2.6 mm thickness in adult males) and connective tissue (ε ≈ 4, σ ≈ 0.25 S/m, about 2.5 mm thickness in adult males). The “skin” tissue in our model is filled using the first material. While this is electromagnetically inaccurate, we preliminarily verified that the impact of this simplification on the final temperature distribution is negligible as the temperature in the first centimeter is mostly determined by the bolus cooling, even when including the (thin) layer of fat.

  1. Page 8, line 252. Please specify in which way the passive antenna is terminated at its input port. Open circuit, short circuit, matched load, other?

We added an explanation of the choice of impedance for the TEM port in Section 2.3 and described how the passive antenna is terminated for this coupling analysis in Section 2.5.3.

  1. Page 9, Eq. (12). The exponent of the n x n matrix is (k-1) or k ? I assume that k is the number of reflections that should be accounted for.

We thank the reviewer for pointing this out. We noticed the mistake in the explanation. “K” is in fact the number of wave “propagations”, which therefore includes the first (forward) excitation. We clarified this in the text.